# Generation and Characterization of Native and Sialic Acid-Deficient IgE

**DOI:** 10.3390/ijms232113455

**Published:** 2022-11-03

**Authors:** Alex J. McCraw, Richard A. Gardner, Anna M. Davies, Daniel I. R. Spencer, Melanie Grandits, Gerd K. Wagner, James M. McDonnell, Sophia N. Karagiannis, Alicia Chenoweth, Silvia Crescioli

**Affiliations:** 1St. John’s Institute of Dermatology, School of Basic & Medical Biosciences, King’s College London, London SE1 9RT, UK; 2Ludger, Ltd., Culham Science Centre, Abingdon, Oxfordshire OX14 3EB, UK; 3Randall Centre for Cell and Molecular Biophysics, School of Basic & Medical Biosciences, King’s College London, London SE1 9RT, UK; 4Medical Biology Centre, School of Pharmacy, Queen’s University Belfast, 97 Lisburn Road, Belfast BT9 7BL, UK; 5Breast Cancer Now Research Unit, School of Cancer & Pharmaceutical Sciences, Guy’s Cancer Centre, King’s College London, London SE1 9RT, UK

**Keywords:** antibodies, immunoglobulin E, IgE, IgE purification, IgE glycosylation, glyco-engineered IgE, neuraminidase, sialic acid, allergy, AllergoOncology

## Abstract

Efficient characterization of IgE antibodies and their glycan structures is required for understanding their function in allergy and in the emerging AllergoOncology field for antibody immunotherapy. We report the generation, glyco-profiling and functional analysis of native and sialic acid-deficient glyco-engineered human IgE. The antibodies produced from human embryonic kidney cells were purified via a human IgE class-specific affinity matrix and structural integrity was confirmed by SDS-PAGE and size-exclusion chromatography (SEC). Purified IgEs specific for the tumor-associated antigens Chondroitin Sulfate Proteoglycan 4 (CSPG4-IgE) and Human Epidermal Growth Factor Receptor 2 (HER2-IgE) were devoid of by-products such as free light chains. Using neuraminidase-A, we generated sialic acid-deficient CSPG4-IgE as example glyco-engineered antibody. Comparative glycan analyses of native and glyco-engineered IgEs by Hydrophilic interaction liquid chromatography (HILIC)-high performance liquid chromatography (HPLC) indicated loss of sialic acid terminal residues and differential glycan profiles. Native and glyco-engineered CSPG4-IgEs recognized Fc receptors on the surface of human FcεRI-expressing rat basophilic leukemia RBL-SX38 cells, and of CD23/FcεRII-expressing human RPMI-8866 B-lymphocytes and bound to CSPG4-expressing A2058 human melanoma cells, confirming Fab-mediated recognition. When cross-linked on the cell surface, both IgEs triggered RBL-SX38 degranulation. We demonstrate efficient generation and functional competence of recombinant native and sialic acid-deficient IgEs.

## 1. Introduction

Immunoglobulin E (IgE) antibodies have long been ascribed well-characterized roles in the pathogenesis of allergy and hypersensitivity [1,2]. In addition, IgE has also been implicated in the immune responses to parasitic worm infections and the clearance of parasites via effector mechanisms [3,4]; and several studies support potential roles of IgE as part of protective immunosurveillance and response against cancer [5,6,7], highlighting a multifaceted immune role for this antibody. More recently, attention has turned towards novel applications of IgE, particularly in a therapeutic setting. Different features of IgE, including a high affinity for FcεRI (although CD23 is low affinity, increased binding can be achieved via avidity [8]); lack of inhibitory Fc receptors or patient Fc receptor polymorphisms; and efficient long-lasting tissue surveillance have earmarked this antibody class as a potential powerhouse in the field of cancer therapeutics [2,9]. This growing interest can be visualized in the nascent field of AllergoOncology, as well as through the recent completion of the Phase I clinical trial of the first-in-class monoclonal IgE antibody (MOv18 IgE) specific for the tumor-associated antigen Folate Receptor alpha (FRα) (NCT02546921) which validated the safety of IgE as a novel cancer treatment as well as demonstrating preliminary anti-tumor efficacy [10].

Separately, interest in IgE glycosylation has been recently reignited. Although heavily glycosylated, with 6 occupied N-linked Fc glycan sites on the constant regions in humans compared to IgG’s sole glycan site [11], limited roles for IgE glycans have been reported with regard to structure and function [12,13]; and, indeed, IgE glycosylation has been largely regarded as non-essential in IgE function as a whole. Only the N394 glycan (the sole oligomannose IgE glycan, considered structurally equivalent to IgG’s Fc glycan) has been implicated in FcεRI binding, although whether this is due to an outright role for this glycan in FcεRI interactions or merely structural stabilization within the IgE molecule has remained controversial [14,15,16,17]. More recently, N394 was conclusively deemed ‘essential’ for induction of IgE responses through FcεRI, with genetic ablation of this glycan found to alter secondary structure and render IgE incapable of binding FcεRI [18]. With regard to complex glycans (representing 5 out of the 6 occupied N-glycan sites [11]), little to no roles have been reported in reference to IgE structure and/or function, and, indeed, they have been ascribed a non-essential role. Supporting this, antibody fragments containing only the N394 glycan have been reported to retain receptor binding and functional activity [15,19,20]. A subsequent study reported increased prevalence of sialic acid residues on IgE from allergic individuals compared to healthy controls, supporting previous observations [21,22], with sialylated IgE driving increased anaphylaxis compared to desialylated IgE [23]. These findings together highlight the need to further evaluate the glycan profiles of IgE, including of IgE sialylation.

Clearer insights into IgE glycosylation will enhance understanding and treatment of allergy and IgE-mediated atopic diseases and will help inform the development of novel antibody scaffolds aimed at filling the gaps in existing monoclonal antibodies for cancer therapy. Consequently, this creates a growing demand for the generation, purification, and characterization of recombinant IgEs with human Fc regions, an endeavor which has historically been highlighted to be an area requiring further research [24,25]. Hence, studies detailing purification of high quality unmodified IgE and production, purification, and characterization of glyco-engineered IgE are highly desired. Existing antibody purification processes predominantly focus on the isolation of IgE from human sera to derive antibodies from relative low serum levels in healthy or allergic individuals, or from disease states such as myeloma or hyper-IgE syndrome [11,17,26,27]. On the contrary, for recombinant IgE generated via cellular expression systems, purification largely involves non-specific matrices which rely on recognition of light chains of any antibody isotype. For specific purification of IgE class antibodies, it is only recently that efforts have been made towards the class specific purification, seeking to address the technical barriers thus far limiting IgE research [24,25]. However, to date, there are no published studies demonstrating the use of class-specific matrices in a laboratory setting for unmodified IgE, or for glyco-engineered IgE.

Herein, we sought to produce, purify, characterize and functionally evaluate matched native and exoglycosidase-generated sialic acid-deficient glyco-engineered versions of IgE. We first evaluated the IgE class-specific affinity matrix for its suitability for small-scale purifications, such as those utilized for glyco-engineering with glycosidase enzymes, versus a non-class specific method. We demonstrated the production and purification of native and glyco-engineered IgE–Neu-IgE, a sialic acid-deficient IgE, building on previous studies investigating sialylation as a contributor to the allergic functions of IgE [23,28]. We studied the structural stability and the glycan structural features of the purified native and sialic acid-deficient glyco-engineered IgE. We evaluated binding to target antigen and Fc receptor-expressing cells, and we conducted preliminary functional assessments of these native and glyco-engineered IgEs. Our findings contribute towards addressing current limitations in the field of recombinant IgE glyco-engineering, glyco-profiling and functional study.

## 2. Results

### 2.1. High Purity of Full-Length IgE Antibody Yields via IgE Class-Specific Affinity Chromatography

Using affinity matrices, we evaluated the purification of two recombinant IgE antibodies: the mouse/human chimeric antibody CSPG4-IgE recognizing the melanoma-associated antigen Chondroitin Sulfate Proteoglycan 4 and the humanized antibody HER2-IgE recognizing the breast cancer-associated antigen Human Epidermal Growth Factor Receptor 2. CSPG4-IgE was generated from stably transfected Expi293F cells as we described previously [29]. HER2-IgE was generated from transiently transfected Expi293F cells, as we previously reported [30]. Both antibodies were originally purified via existing methods using light chain (e.g., HiTrap™ KappaSelect columns, herein referred to as KappaSelect). We used CSPG4-IgE as a benchmark to validate the purification of IgE in a laboratory setting using an IgE class-specific affinity matrix (CaptureSelect^TM^ IgE Affinity Matrix, herein referred to as CaptureSelect™) and assessed its suitability for small-scale purifications and downstream glyco-engineering.

First, using IgE class-specific affinity matrix-isolated CSPG4-IgE, we evaluated the elution conditions able to generate the highest antibody yields from culture supernatants using Pierce™ Micro-Spin Columns (ThermoFisher Scientific, Waltham, MA, USA, Cat. 89879). We assessed packing conditions using 100, 200, 300, or 400 μL of IgE class-specific matrix slurry in 500 μL spin columns [Figure 1A,B], looking for consistent bed heights and return of known volumes of IgE. Using 50 μg CSPG4-IgE, we confirmed that, although 100 μL slurry is recommended by manufacturers, there is no significant difference in percentage yield recovery across the different resin volumes, which averaged at approximately 80% return of antibody [Figure 1A], potentially allowing for purification of higher concentrations of antibody using a small purification set-up. SDS-PAGE analysis under non-reducing conditions confirmed the purification of full length IgE at the expected molecular weight. No antibody was detected in the flow-through, suggesting that all IgE product was purified using variable resin volumes [Figure 1B]. We selected a 200 μL resin volume for future studies. With the IgE class-specific affinity matrix offering a binding capacity of >5 mg/mL, this volume should provide the capacity to purify up to 1 mg antibody.

We next investigated the impact of different elution buffers on antibody yields returned following purification, using three elution buffers: Buffer 1, 50 mM sodium citrate and 150 mM NaCl (pH 3.5) (recommended by the manufacturer); Buffer 2, 0.1 M glycine (pH 2.3) (used for light chain-specific purification, and other affinity matrices); and Buffer 3, 20 mM citric acid (pH 3.0), another common purification buffer, used for purification in our laboratory [Figure 1C,D]. Whilst all 3 buffers were capable of eluting IgE, only Buffer 1 gave consistently high returns of a known volume of IgE (100 μg) from the matrix in four independent experiments [Figure 1C]. SDS-PAGE under non-reducing conditions was used to confirm purification of full-length IgE, with all IgE product appearing in the elute and not in the flow-through, suggesting purification of antibodies with each buffer [Figure 1D].

Using a 200 μL resin slurry volume, micro-spin columns and Buffer 1 for IgE purification, we contrasted the IgE class-specific matrix with the kappa light chain-specific matrix previously used for purification of IgEs in previous studies [29]. We conducted these comparisons for the purification of two IgE antibodies with human Fc regions, namely the chimeric CSPG4-IgE and the humanized trastuzumab equivalent HER2-IgE, both generated in Expi293F cells as we describe above. For both antibodies, we observed IgEs of higher purity using IgE class-specific (CS) compared to our previously reported standard light chain-specific (KS) purification matrix. Free light chain and other impurities, indicated in the KS-purified product, were absent in both CS-purified IgE samples as visualized by SDS-PAGE [Figure 1E]. Similarly, size-exclusion chromatography (SEC) analysis indicated superior purity and absence of by-products for both IgEs purified using IgE class-specific isolation [Figure 1F,G], which showed the presence of solely intact antibody in both CS-purified products. SEC analyses showed that lower molecular weight degradation products, visible at 30–35 min for CSPG4-IgE [Figure 1F] and between 20–30 min for HER2-IgE [Figure 1G], were completely absent from the final IgE preparation when purified via IgE class-specific affinity and there was no significant presence of aggregation or degradation. Flow cytometry studies indicated comparable binding profiles of both light chain-purified (KS-IgE) and IgE class-specific affinity-purified CSPG4-IgE (CS-IgE) to rat basophilic leukemia cell line RBL-SX38 cells expressing the high-affinity human IgE receptor FcεRI [Figure 1H].

### 2.2. Generation and Purification of Sialic Acid-Deficient IgE

We next wished to generate glyco-engineered IgE, via glycosidase. Previous reports suggested that sialic acid glycans on IgE may participate in allergic functions [23]. We therefore designed a pipeline for the removal of sialic acid. For this we selected neuraminidase-A (Neu), a broad specificity exoglycosidase targeting both branched and linear terminal sialic acid residues, to produce CSPG4-IgE with reduced sialic acid (Neu-IgE). We then compared this variant against native CSPG4-IgE (Con-IgE). CSPG4-IgE was incubated with Neu and the antibody was then purified using IgE class-specific affinity chromatography.

We first wished to evaluate whether enzyme incubation conditions (in the absence of the enzyme), such as the low pH of the GlycoBuffer (pH 5.5 compared to physiological pH of approximately 7), could impact IgE interactions with its Fc receptors independently of the potential impact of neuraminidase-A. We therefore evaluated IgE binding to the high affinity FcεRI receptor expressed on RBL-SX38 cells and to the low affinity receptor CD23/FcεRII expressed on human B lymphoblastoid RPMI-8866 cells by flow cytometry. We observed that binding of antibodies at different concentrations to these cells was similar for CSPG4-IgE incubated with GlycoBuffer for 16 h at 37 °C and CSPG4-IgE incubated in PBS alone [Figure 2A,B]. This suggested that incubation with GlycoBuffer alone (in the absence of Neuraminidase-A) did not appear to adversely affect the ability of IgE to bind to either one of its cognate receptors on immune cells.

We then prepared and purified Neu-IgE using IgE class-specific affinity chromatography as described above. We confirmed removal of neuraminidase from the final preparation (Neu-IgE), as indicated by green arrows [Figure 2C]. Next, we assessed unmodified Con-IgE and Neu-IgE produced and purified via IgE class-specific affinity chromatography under both non-reducing and reducing conditions [Figure 2D] to confirm structural integrity of both antibodies and lack of impurities. We additionally tested the purification of Neu-IgE treated with neuraminidase for either 2 h or 16 h to investigate any potential impact of an extended incubation time in case these conditions had adverse effects on antibody stability. SDS-PAGE analyses under non-reducing conditions comparing full-length IgE incubated with neuraminidase for either 2 h or 16 h at 37 °C against an untreated control showed no differences. Furthermore, when 2 h and 16 h Neu-IgE were compared under reducing and non-reducing conditions to investigate for potential differences between preparations, we observed no significant changes in band size [Figure 2E]. Finally, we assessed the structural integrity of IgE class-specific affinity-purified Neu-IgE preparation via SEC analysis [Figure 2F]. This demonstrated that Neu-IgE (blue) showed a similar retention time and SEC profile to that of Con-IgE (red). Together, these suggest the generation of full-length IgE at high purity following neuraminidase treatment, with no significant degradation or aggregation.

### 2.3. Glyco-Analysis Retention of Complex Glycan Structures and Loss of Sialic Acid Residus on Glyco-Engineered IgE 

We next evaluated the glycan profiles of matched native (Con-IgE) versus sialic acid-deficient (Neu-IgE) antibodies specific for CSPG4. This was conducted via release of the N-glycan structures from both Con-IgE and Neu-IgE using PNGase F, procainamide labelling of the released N-glycan structures and separation and detection of the antibody glycan profiles via high-performance liquid chromatography fluorogenic derivatization mass spectrometry HPLC-FD-MS [31]. HPLC-FD-MS profiling of both Con-IgE and Neu-IgE and comparison of chromatograms and MS profiles confirmed complete removal of terminal sialic acid structures from the IgE glycans. The most readily observed indicator of this was the loss of peak numbers 46–55 in the Neu-IgE chromatogram between 35 and 41 min [Figure 3, bottom panel] compared to Con-IgE [Figure 3, top panel, IgE incubated with GlycoBuffer alone]. These multi-sialylated, tetra-antennary glycan structures were not present and had been digested with the neuraminidase-A resulting in an increase in peak 43, a tetra-galactosylated, tetra-antennary glycan, and in peaks 36–38, tri-galactosylated, tetra-antennary glycans. Chromatogram comparisons also show that Neu-IgE is much less complex than Con-IgE due to the loss of sialic acids showing the impact that these structures have on glycan complexity. 

Analysis of the Mass Spectrometry (MS) profiles of Neu-IgE and Con-IgE resulted in a comprehensive list of predicted monosaccharide compositions and suggested glycan structures for each peak [Table 1 and Table 2]. MS analysis showed that there were no peaks detected in Neu-IgE that contained sialylation [Table 2]. The absence of sialylation was confirmed by obtaining Extracted Ion Chromatograms (EIC) from the MS for Neu-IgE for two MS fragment ions containing sialic acid, Neu5Ac-Gal-GlcNAc (657.34 ± 0.1) and Neu5Ac-GalNAc-GlcNAc (698.34 ± 0.1). Neither of these fragment ions were found in the MS for Neu-IgE, showing that none of the detected structures contained sialic acid. In contrast, a number of both fragment ions were found in the MS for Con-IgE [Appendix A]. Except for the sialylation, Neu-IgE retained its glycan structures in a manner comparable with Con-IgE. These findings confirmed that IgE glycan structures were not adversely affected, beyond the loss of sialic acid which was complete.

### 2.4. Glyco-Engineered IgE retains Fc and Fab Region-Mediated Binding to Immune and Target Antigen Expressing Cells and can Trigger Cellular Degranulation

We assessed whether Neu-IgE could recognize Fc receptor-expressing and target antigen-expressing cells via its Fc and Fab regions, respectively. The Fc-mediated properties of Neu-IgE were examined by investigating recognition of FcεRI expressing RBL-SX38 cells and recognition of CD23-expressing RMPI-8866 cells. Neu-IgE was able bind human FcεRI-expressing cells in a dose-dependent manner comparable to that of native CSPG4-IgE (Con-IgE) [Figure 4A]. Neu-IgE appeared to show heightened ability to bind to CD23 compared to the Con-IgE control [Figure 4B]. Both Con-IgE and Neu-IgE recognized and bound the target antigen CSPG4-expressing A2058 human melanoma cells in a similar dose-dependent manner [Figure 4C], suggesting that sialic acid reduction had no obvious impact on the Fab-mediated binding of IgE on target antigen-expressing cancer cells. When cross-linked using a polyclonal anti-IgE antibody, both Con-IgE and Neu-IgE were able to trigger degranulation of RBL-SX38 cells [Figure 4D]. These findings suggest that glyco-engineered IgEs retained Fc and Fab region-mediated binding to immune and target antigen expressing cells and demonstrated functional capability. 

## 3. Discussion

For studying the roles of IgE in Allergy and AllergoOncology, as well as for the purpose of investigating potential therapeutic applications of IgE antibodies for the treatment of cancer, it is important to develop efficient and reproducible pipelines for the production of native and glyco-modified IgEs, and to evaluate the biophysical and biological characteristics of IgE glyco-variants. Previously, we reported the stable production of recombinant IgE in mammalian human embryonic kidney (Expi293F) cells, which we exemplified using a melanoma-associated antigen (CSPG4)-specific IgE antibody [29]. We wished to build and improve upon these processes, both aiming to achieve yields of high-purity IgE and for the purpose of IgE antibody glyco-engineering, glyco-profiling, characterization and functional evaluations. Our findings suggest that human IgE class-specific affinity chromatography can be employed for the isolation of glyco-engineered IgEs in a small-scale setting. Using a CSPG4-specific IgE generated from stably transfected Expi293F cells, we demonstrate improved purification via an IgE class-specific affinity matrix, superior to previously reported chromatography approach which utilizes kappa light chain recognition. Native and sialic acid-deficient IgEs showed comparable purity, structural and functional integrity [Figure 1]. IgE production and characterization in the fields of Allergy and AllergoOncology are highly desirable, especially as interest continues to grow surrounding the functional attributes of IgE antibodies in different disease settings and the potential roles of glycans on the IgE structure. The use of a global human IgE class-specific affinity chromatography for the production of glycoengineered IgE has not previously been demonstrated in the literature. With regard to generating a glycan-modified IgE by using enzymatic digestion, we showed that the enzyme incubation conditions, such as a lower pH of 5.5, did not adversely affect IgE structure, stability, or ability to bind to FcεRI- and CD23-expressing cells. This would have been potential concern due to the extended period of time antibodies may be kept outside of physiological conditions for enzymatic digestion. Furthermore, we confirmed the removal of impurities and additives such as enzymes for the small-scale glyco-engineered IgE preparation, and we showed that our sialic acid-deficient IgE remained intact as compared to the native IgE equivalent [Figure 1 and Figure 2]. Our HILIC-HPLC analyses confirmed desialylation, whilst the remaining native glycan structures on the antibody appeared to have been preserved [Figure 3]. Sialic acid-deficient IgE was devoid of multi-sialylated, tetra-antennary glycan structures resulting in Neu-IgE showing lower glycan complexity compared with Con-IgE [Figure 4, Table 1 and Table 2]. Despite the loss of sialic acid residues, we showed that glyco-engineered IgE produced and purified via this pipeline retained recognition of immune cells expressing both cognate IgE Fc receptors, FcεRI and CD23, as well as of cancer cells expressing the target antigen, CSPG4 [Figure 4]. Furthermore, glyco-engineered IgE was able to drive degranulation of RBL-SX38 cells on par with native IgE (Con-IgE) and an IgE isotype positive control when the antibodies were cross-linked with polyclonal anti-IgE. These data showcase the production, purification, structural and functional assessment of a stable glyco-engineered IgE via IgE class-specific affinity chromatography (outlined in Appendix A).

It has long been known that differential glycosylation is common in IgG antibody isotypes, and that the presence or absence of certain glycan features can influence the functional capabilities of IgG [32,33,34,35]. For example, terminal Fc sialylation has been linked to the anti-inflammatory activities arising from intravenous immunoglobulin therapy [36], while core fucosylation is well described in the moderation of Fc-mediated effector functions such as antibody-dependent cellular cytotoxicity (ADCC) of IgG, by interfering with its ability to interact with various Fcγ receptors [37,38,39]. Similarly, certain disease states such as rheumatoid arthritis [40], systemic lupus erythematosus, Crohn’s disease [41] and autoimmune thyroid diseases (AITD) [42] show patterns of differential glycosylation on circulating IgGs. Such observations have informed the development of therapeutic IgG antibodies [43]. With growing interest in IgE and its roles not only in allergic diseases, but also as a novel alternative therapeutic antibody class for cancer therapy, the generation of purified IgEs and the evaluation of their glycan profiles is still required. The launch of a Phase I clinical trial of the first-in-class monoclonal IgE antibody (MOv18 IgE) specific for the tumor-associated antigen Folate Receptor alpha (FRα), which was recently completed (NCT02546921) [10], highlights the importance of developing efficient pipelines for recombinant IgE production, purification, characterization and glycan analysis.

Investigating the role of IgE glycan functions has been complicated due to the presence of seven N-linked glycosylation sites of which six are occupied [11,28], as compared to only one site in the IgG Fc region. Although knowledge surrounding the function of these IgE glycans is relatively limited, recent observations have linked terminal Fc sialylation in the human IgE antibody to allergic pathogenicity [23]. Examination of total IgE derived from peanut-allergic individuals and healthy volunteers showed increased sialic acid content on the IgE derived from allergic individuals. Subsequent removal of sialic acid was able to successfully attenuate effector cell-driven degranulation across a multitude of functional models as well as to reduce anaphylaxis [23]. These findings suggested that IgE sialylation may be a regulator of allergic disease; and previous work has similarly demonstrated evidence for roles of sialic acid in modulating IgE function [21,22]. Thus, we prioritized sialic acid-deficient IgE as our representative glyco-engineered variant. Interestingly, our sialic acid-deficient IgE demonstrated increased binding to the low-affinity CD23 receptor, but not to the high-affinity FcεRI [Figure 4B]. This data support previous work on *in planta* glyco-engineered IgEs, in which we demonstrated the ability of the IgE glycan profile to modulate the binding to CD23-expressing [28], but not to FcεRI-expressing cells, which will be of interest to investigate in future work. In keeping with comparable binding to FcεRI, there was no discernible difference in the ability of sialic acid-deficient IgE to recognize or trigger functional degranulation of FcεRI-expressing RBL-SX38 cells.

Our findings of the generation of intact, functionally active IgE antibodies with defined glycan profiles address a significant gap in the IgE field and will facilitate ongoing research into understanding the roles of IgE glycosylation in health and disease. We demonstrate a reliable means of producing and purifying both native and glyco-engineered IgE antibodies using a human IgE class-specific affinity matrix which we illustrate here by the generation of sialic acid-deficient IgE, on a small-scale, with easy scalability for larger-scale purifications, including the purification of IgE directly from cell culture supernatants. Whilst we chose to use neuraminidase-treated IgE as our representative glyco-engineered antibody due to increasing interest in IgE-associated sialic acid residues, this pipeline can be amenable for purification of other glyco-engineered antibodies, including those produced with glycosidases and through genetic engineering. Although not shown here, we have also successfully used IgE class-specific affinity chromatography the purification of IgE antibodies from cell culture supernatants, including from cultures co-treated with glycosyltransferase inhibitors.

In conclusion, we have generated and evaluated native and sialic acid-deficient IgEs with human Fc regions. The antibodies showed high purity compared to those isolated via conventional means such as with light chain-based purification methods. We have shown that the glyco-engineering process had no adverse effects on the basic structure and cell binding attributes of the IgE antibodies such as Fc receptor binding and antigen recognition. We present a full characterization of paired sialic acid-deficient and native IgE variants including structural and glycan profile comparisons as well as functionality in cell-based assays. Our study may address the current need in the field of IgE biology and represents an accessible, adaptable, and reproducible means of rapidly generating, characterizing and functionally evaluating native and glyco-engineered IgE antibodies.

## 4. Materials and Methods

### 4.1. Production of Recombinant IgE in Culture Using Human Embryonic Kidney Expi293F Cells

Expi293F cells were stably transfected to express IgE specific for the melanoma-associated antigen Chondroitin Sulfate Proteoglycan 4 (CSPG4) [29] and Expi293F cells were transiently transfected to express HER2-specific IgE [30]. Cells were seeded in 125 mL Erlenmeyer Flasks [SLS, Nottingham, UK, Cat. 431143] at cell densities of 5 × 10^6^ cells/mL and incubated for 3 days under shaking conditions. On the third day after seeding, cells were counted, and supernatants were harvested and filtered twice through 0.45 μm and 0.20 μm filters prior to antibody purification. 

### 4.2. Packing of Chromatography Columns and Purification of Recombinant IgE from Culture Supernatants

A C10/10 column [Cytiva, Marlborough, MA, USA, Cat. 19500101] was assembled according to manufacturers’ instructions and connected to a P-1 peristaltic pump [Cytiva, Marlborough, MA, USA, Cat. 18111091]. CaptureSelect^TM^ IgE Affinity Matrix [ThermoFisher Scientific, Waltham, MA, USA, Cat. 2943542005] was gently resuspended, then 5 ml of resin transferred into the column. The column was fitted with an AC-10 Flow Adapter [Cytiva, Marlborough, MA, USA, Cat. 19500601] to allow for a smaller resin bed. Prior to use, the column was flushed with at least 3 column volumes (CV) to equilibrate following constant bed height. Following resin equilibration with PBS, cell supernatants were diluted 1:1 with PBS and passed through the column at a rate of 5 ml/minute. Resin was then washed again with PBS. Captured IgE was eluted using 20 mM citric acid (pH 3.0) or 50 mM sodium citrate + 50 mM sodium chloride (pH 3.5) [Table 3] and neutralized with 1 M Tris (pH 8.2) for a total of 10 1 ml aliquots. Resin was first washed with 3 CV 20 mM citric acid, then with 3 CV 20% ethanol. A microscale UV-Vis spectrophotometer [Nanodrop ND-1000, Labtech International Ltd, Heathfield, UK] was used to determine fractions containing antibody.

Purification via light chain specific affinity chromatography was performed using HiTrap™ KappaSelect column [Cytiva, Marlborough, MA, USA, Cat. 17545812] according to manufacturer’s instructions.

### 4.3. Dialysis of Recombinant IgE from Culture

For bulk purification, fractions containing antibody product were pooled to a maximum volume of 6 ml. 4 L of PBS was prepared, and samples transferred to Tube-O-DIALYZER Medi-50kDa dialysis tubes [G-Biosciences, St. Louis, MO, USA, Cat. 786.619] floated in PBS on a magnetic plate with a stirrer and left to dialyze overnight at 4 °C. For fractions totaling less than 1 ml, fractions were transferred instead to Slide-A-Lyzer Dialysis Casettes (3.5k MWCO, 0.5 ml) [ThermoFisher, Waltham, MA, USA, Cat. 66333] with PBS and dialyzed for 2 h at 4 °C. After 2 h, PBS was changed, then samples incubated overnight with stirring at 4 °C. Sample concentrations were measured via a spectrophotometer as described and concentrated if needed via Amicon Ultra-4 Centrifugal 50 Kda Filters [Merck-Millipore, Burlington, VT, USA, Cat. UFC805008]. All samples were stored at 4 °C following purification and dialysis.

### 4.4. Production and Purification of Sialic Acid-Deficient IgE using Glycosidase Enzymes

Unmodified IgE was incubated with A2-3,6,8,9 Neuraminidase [New England BioLabs, Ipswich, SD, USA, Cat. P0722L] for 2 h at 37 °C. Neuraminidase-A is a broad specificity exoglycosidase capable of cleaving both branched and linear terminal sialic acids with α2-3, α2-6, α2-8, and α2-9 linkages. Pierce™ Micro-Spin Columns [Thermo-Fisher Scientific, Waltham, MA, USA, Cat. 89879] were packed with 100 μL CaptureSelect™ IgE Affinity Matrix and resin equilibrated with PBS. Columns were capped at the base, sample loaded to a maximum volume of 400 μL with PBS then capped and incubated for 30 mins at RT with end-over-end mixing. Columns were placed in 2 ml collection tubes and centrifuged for 1 min at 10,000× *g* then washed for 3 CV with PBS. Bound IgE was eluted [Table 3, Buffers 1–3] and neutralized with 1 M Tris (pH 8.2) for a total of five 220 μL fractions. To restore columns, resin was washed with elution buffer and 20% ethanol for 3 CV each and stored at 4 °C in 20% ethanol. Samples were measured via nanodrop, pooled and dialyzed as described above.

### 4.5. Confirmation of Antibody Purification via Sodium Dodecyl Sulphate Polyacrylamide Gel Electrophoresis (SDS-PAGE)

Antibody purification was confirmed using SDS-PAGE. Samples were mixed with Laemmli Buffer 4× [Bio-Rad, Hercules, CA, USA, Cat. #1610747] or boiled at 95 °C for 5 mins with Reducing Buffer [Table 2] for reducing conditions then loaded into Mini-PROTEAN TGX Gels, (15-well, 15 μL) [Bio-Rad, Hercules, CA, USA, Cat. #4561036]. Gels were run at 150 V for 45 mins on a Mini-PROTEAN Tetra Vertical Electrophoresis Cell [Bio-Rad, Cat. #1658004] and visualized using InstantBlue Protein Stain [Sigma-Aldrich, St. Louis, MO, USA, Cat. ISB1L]. 

### 4.6. Size Exclusion Chromatography (SEC)

The samples were filtered through a 0.2 μm filter immediately before the experiment. Size exclusion chromatography analysis was performed using a Superdex 200 column that had been previously equilibrated with PBS containing 0.1% (*w*/*v*) sodium azide. 

### 4.7. Glycoanalysis of Con-IgE and Neu-IgE by Hydrophobic Interaction Liquid Chromatography (HILIC) HPLC

Samples were used as supplied, with no clean up, but dried down before use. Samples were treated with PNGase F to release the N-glycans and then cleaned up prior to procainamide labelling. Following the labelling, the samples were cleaned up further to remove excess reagents, eluted in water from the cleanup plate and concentrated prior to HPLC-FD-MS analysis as described previously [31]. Samples were separated and analyzed via Hydrophilic Interaction Liquid Chromatography (HILIC) HPLC using a Dionex Ultimate 3000 UHPLC instrument using a BEH-Glycan 1.7 µm, 2.1 × 150 mm column (Waters) with a fluorescence detector (λex = 310 nm, λem = 370 nm) controlled by Bruker HyStar 3.2 and Chromeleon data software version 7.2. MS analysis was performed using a Bruker mazon Speed ETD electrospray mass spectrometer, which was coupled directly after the UHPLC FD without splitting. HPLC-ESI-MS chromatogram analysis was performed using Bruker Compass DataAnalysis 4.4 and GlycoWorkbench software. Chromeleon Data software, version 7.2, was used to allocate glucose unit (GU) values to peaks.

### 4.8. Flow Cytometric Evaluations of IgE Binding to Cell Surface Receptors and Antigens

Antibody binding to cell surface Fc receptors FcεRI and CD23 and to the tumor-associated antigen CSPG4 was analyzed via flow cytometry. FcεRI-expressing and CSPG4-expressing adherent cells were detached using 0.5 mM EDTA, resuspended in FACS buffer [Table 3] and incubated at 4 °C with serially diluted CSPG4-IgE or Neu-IgE for 30 mins in FACS tubes. Cells were washed with PBS, then incubated on ice for 20 mins with anti-IgE-Fluorescein [Vector Laboratories, Burlingame, CA, USA, Cat. FI-3040] or anti-IgE-APC [BioLegend, San Diego, CA, USA, Cat. 325508] and washed again with PBS. Cells were resuspended in FACS buffer prior to analysis. A similar protocol was used for analyzing CD23 binding on RPMI-8866 cells, with RPMI-1640 medium + 2% FBS (fetal bovine serum) being used in place of FACS buffer to provide higher Ca^2+^ concentration. Samples were analyzed on a FACS Canto II [BD Biosciences] and results analyzed using FlowJo v10.8.1 software. Data was analyzed on GraphPad Prism 9 and non-linear regression curve fits used to calculate the EC50.

### 4.9. IgE-Mediated Degranulation of RBL-SX38 Cells

RBL-SX38 cell degranulation was measured by quantifying release of β-hexosaminidase, as described previously [29]. Cells were seeded at 1 × 10^4^ cells/well overnight in culture medium and the next day, sensitized with 200 ng/mL IgE, control antibody, or medium alone by incubating for 1 h at 37 °C. Cells were washed 3 times with stimulation buffer (HBSS + 2% FBS) and stimulated for 1 h at 37 °C with either stimulation buffer alone or rabbit anti-IgE (1.5 μg/mL). For quantifying β-hexosaminidase, 25 μL culture supernatant was diluted 1:1 with stimulation buffer and transferred onto black 96-well plates containing 50 μL florigenic substrate per well (1 mmol/L 4-methyllumbelliferyl N-acetyl b-D-glucosaminide, 0.1% dimethyl sulfoxide, 200 mmol/L sodium citrate, pH 4.5). Plates were incubated for 1 h in the dark at 37 °C then the reaction quenched with 100 μL per well of 0.5 M Tris. Plates were read with a FLUOstar Omega Microplate Reader (350-nm excitation, 450-nm emission; BMG Labtech, Ortenberg, Germany). Degranulation was expressed as a percentage of Triton X-100 release (100%).

### 4.10. Statistical Analysis

Error bars represent SDs or SEMs. Statistical significance of degranulation assays was calculated using 1-way ANOVA with the Tukey’s multiple comparisons test. *p* values of less than 0.05 were considered significant. Data were analyzed using GraphPad Prism 9 software. 

## Figures and Tables

**Figure 1 ijms-23-13455-f001:**
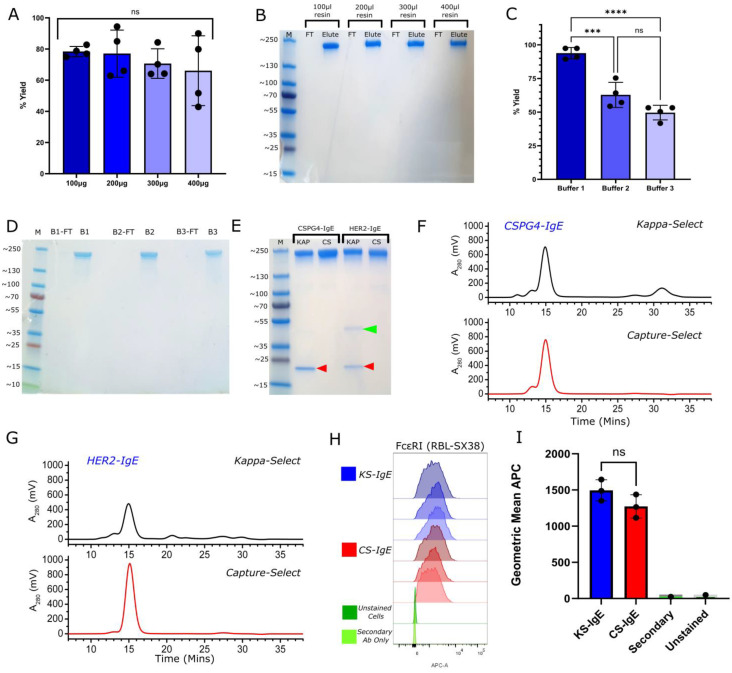
Optimization of IgE Purification using IgE class-specific affinity chromatography. (**A**). Quantitation of the % yield CSPG4-IgE antibody recovered using varying volumes of resin slurry from a set IgE volume of 100 μg per purification, N = 4, error bars = SD (Standard Deviation); (**B**)**.** InstantBlue-stained SDS-PAGE of IgE purified using variable resin volumes against a set concentration of IgE under non-reducing conditions, representative experiment from A (FT: flow through; Elute: column elute; M: molecular marker); (**C**)**.** Quantitation of the % yield of CSPG4-IgE recovered using different elution buffers from a set IgE volume of 100 μg and a resin slurry volume of 200 μL. Buffer 1 = 50 mM Sodium Citrate + 50 mM NaCl, pH 3.5; Buffer 2 = 0.1 M Glycine, pH 2.3; Buffer 3 = 20 mM Citric Acid, pH 3.0. N = 4, error bars = SD; *** *p* > 0.005 **** *p* > 0.001 (**D**). InstantBlue-stained SDS-PAGE of IgE purified using different elution buffers against a set concentration of IgE, shown under non-reducing conditions. B1: Buffer 1 = 50 mM Sodium Citrate + 50 mM NaCl, pH 3.5; B2: Buffer 2 = 0.1 M Glycine, pH 2.3; B3: Buffer 3 = 20 mM Citric Acid, pH 3.0. Representative independent experiment from C (B-FT: flowthrough; B: column elute; M: molecular marker); (**E**)**.** InstantBlue-stained SDS-PAGE of IgE purified using light chain-specific (KappaSelect) or IgE class-specific (CaptureSelect™) affinity chromatography, for CSPG4-specific IgE and HER2-specific IgE. M, molecular marker; KAP = KappaSelect-purified; CS = CaptureSelect™-purified. Red arrows indicate free light chain; green arrows indicate additional impurities in KS-purified HER2-IgE; (**F**). SEC analysis of CSPG4-specific IgE purified via light chain (black) versus IgE class-specific (red) affinity chromatography; (**G**)**.** SEC analysis of HER2-specific IgE purified via light chain (black) versus IgE class-specific (red) affinity; (**H**). Flow cytometric histograms of CSPG4-IgE purified via either light chain (Kappa-Select, KS-IgE) [N = 3, blue, triplicates] or IgE class-specific (CaptureSelect™) (CS-IgE) [N = 3, red, triplicates] matrices, showing binding of 50 μg/mL IgE to FcεRI on RBL-SX38 cells. Unstained cells and secondary control are shown in green, N = 1; (**I**). Geometric Means of KS-IgE [N = 3] and CS-IgE [N = 3] for binding shown in H.

**Figure 2 ijms-23-13455-f002:**
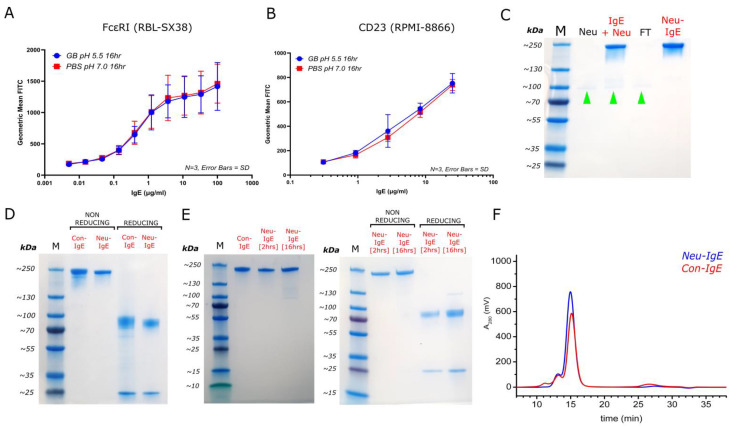
Small-scale generation of sialic acid-deficient CSPG4-IgE (**A**). Effects of enzyme incubation conditions on IgE binding activity to FcεRI on RBL-SX38 cells. GB = GlycoBuffer 1, pH 5.5; PBS = PBS, pH 7.0. N = 3, error bars = SD; (**B**). Effects of enzyme incubation conditions on IgE binding activity to CD23 on RPMI-8866 cells. GB = GlycoBuffer 1, pH 5.5; PBS = PBS, pH 7. N = 3, error bars = SD; (**C**). InstantBlue-stained SDS-PAGE to confirm purification of Neu-IgE via IgE class-specific affinity chromatography M, molecular marker; Neu, Neuraminidase-only control sample; IgE+ Neu, post-neuraminidase incubation sample containing CSPG4-specific IgE + neuraminidase; FT, affinity chromatography flowthrough sample; Neu-IgE, neuraminidase-treated IgE post-purification. Green arrows indicate location of neuraminidase; (**D**). InstantBlue-stained SDS-PAGE comparing Con-IgE and Neu-IgE under non-reducing and reducing conditions; (**E**). InstantBlue-stained SDS-PAGE showing Neu-IgE incubated at either 2 h or 16 h in comparison to Con-IgE under non-reducing conditions and comparing Neu-IgE (2 h) vs. Neu-IgE (16 h) under non-reducing and reducing conditions; (**F**). SEC analysis of Neu-IgE (blue) in comparison with unmodified (native) CSPG4-IgE (Con-IgE, red) both purified via IgE class-specific affinity chromatography.

**Figure 3 ijms-23-13455-f003:**
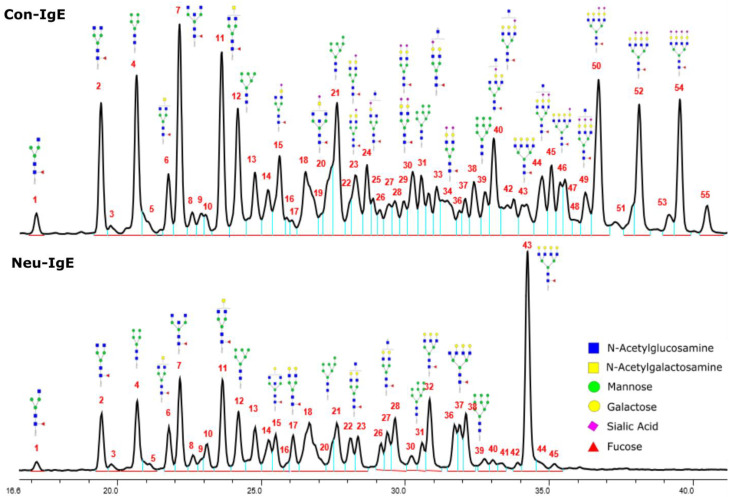
HPLC-FD chromatograms for Con-IgE and Neu-IgE, with suggested glycan structures assigned to main peaks based on m/z masses and predicted monosaccharide compositions. Observed m/z with predicted monosaccharide compositions and suggested structures for each labelled peak are shown in Table 1 [Con-IgE] and Table 2 [Neu-IgE].

**Figure 4 ijms-23-13455-f004:**
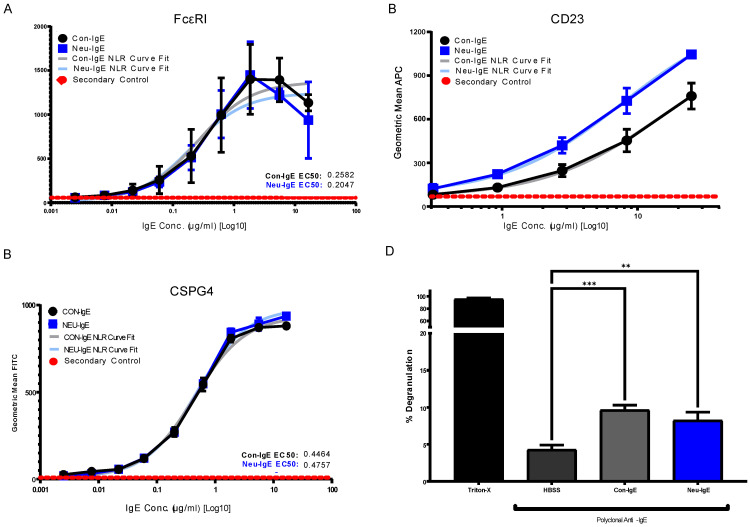
Binding of Neu-IgE and Con-IgE on Fc receptor and target antigen-expressing cells. (**A**). Binding to FcεRI-expressing RBL-SX38 cells by Con-IgE and Neu-IgE with EC50, N = 3, error bars = SD; (**B**). Binding to CD23-expressing RPMI-8866 cells by Con-IgE and Neu-IgE, N = 3, error bars = SD; (**C**). Binding to CSPG4-expressing (target antigen) A2058 cells by Con-IgE and Neu-IgE with EC50, N = 3, error bars = SD; (**D**). Degranulation of RBL-SX38 cells. Data represent the mean ± SEM of 9 independent experiments. ** *p* < 0.01, *** *p ≤* 0.001.

**Table 1 ijms-23-13455-t001:** Predicted Monosaccharide Compositions and Suggested Glycan Structures for Con-IgE based on observed MS m/z masses. Numbered peaks are shown in Figure 3. H = Hexose; N = N-Acetylhexosamine; F = Fucose; S = Sialic Acid.

Peak Number	Observed MS	Calculated MS	Predicted Monosaccharide Composition	Suggested Structure
1	740.42 ^2+^	740.33 ^2+^	H3N3F1-PROC	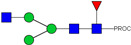
2, 3	841.95 ^2+^	841.87 ^2+^	H3N4F1-PROC	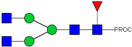
4, 5	727.90 ^2+^	727.81 ^2+^	H5N2-PROC	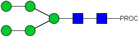
6	943.50 ^2+^	943.41 ^2+^	H3N5F1-PROC	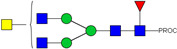
7	943.50 ^2+^	943.41 ^2+^	H3N5F1-PROC	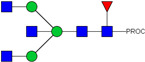
8	922.98 ^2+^	922.89 ^2+^	H4N4F1-PROC	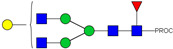
9	914.97 ^2+^	914.89 ^2+^	H3N4F2-PROC	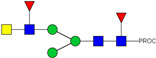
10	922.98 ^2+^	922.89 ^2+^	H4N4F1-PROC	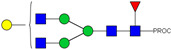
11	943.50 ^2+^	943.41 ^2+^	H3N5F1-PROC	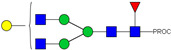
	1045.05 ^2+^	1044.94 ^2+^	H3N6F1-PROC	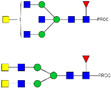
12	808.92 ^2+^	808.84 ^2+^	H6N2-PROC	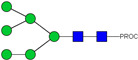
13	683.37 ^3+^	683.29 ^3+^	H4N5F1-PROC	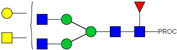
	678.03 ^3+^	677.96 ^3+^	H3N5F2-PROC	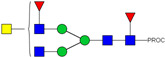
14	683.36 ^3+^	683.29 ^3+^	H4N5F1-PROC	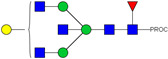
15	683.37 ^3+^	683.29 ^3+^	H4N5F1-PROC	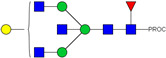
	726.39 ^3+^	726.30 ^3+^	H3N5F1S1-PROC	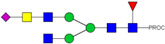
16	996.02 ^2+^	995.92 ^2+^	H4N4F2-PROC	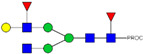
17	669.68 ^3+^	669.61 ^3+^	H5N4F1	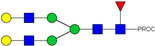
18	1024.50 ^2+^	1024.43 ^2+^	H4N5F1-PROC	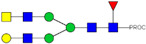
	745.73 ^3+^	745.65 ^3+^	H3N6F2-PROC	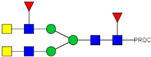
	751.06 ^3+^	750.98 ^3+^	H4N6F1-PROC	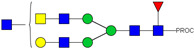
19	712.72 ^3+^	712.63 ^3+^	H4N4F1S1-PROC	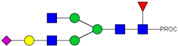
20	794.07 ^3+^	794.00 ^3+^	H3N6F1S1-PROC	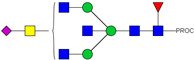
21	889.94 ^2+^	889.86 ^2+^	H7N2-PROC	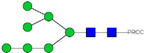
22	737.43 ^3+^	737.31 ^3+^	H5N5F1-PROC	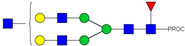
23	737.43 ^3+^	737.31 ^3+^	H5N5F1-PROC	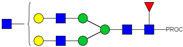
	766.72 ^3+^	766.65 ^3+^	H5N4F1S1-PROC	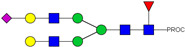
	780.42 ^3+^	780.32 ^3+^	H4N5F1S1-PROC	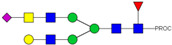
24, 25	848.10 ^3+^	848.02 ^3+^	H4N6F1S1-PROC	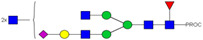
26	780.41 ^3+^	780.32 ^3+^	H4N5F1S1-PROC	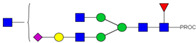
	805.08 ^3+^	805.00 ^3+^	H5N6F1-PROC	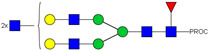
27	805.09 ^3+^	805.00 ^3+^	H5N6F1-PROC	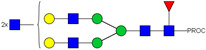
28	766.72 ^3+^	766.65 ^3+^	H5N4F1S1-PROC	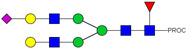
	805.08 ^3+^	805.00 ^3+^	H5N6F1-PROC	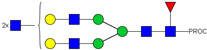
29	834.43 ^3+^	834.34 ^3+^	H5N5F1S1-PROC	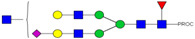
	848.10 ^3+^	848.02 ^3+^	H4N6F1S1-PROC	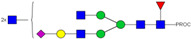
30	834.43 ^3+^	834.34 ^3+^	H5N5F1S1-PROC	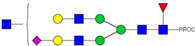
	863.77 ^3+^	863.68 ^3+^	H5N4F1S2-PROC	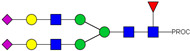
	877.42 ^3+^	877.35 ^3+^	H4N5F1S2-PROC	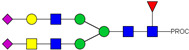
31	970.95 ^2+^	970.89 ^2+^	H8N2-PROC	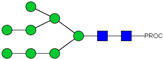
32, 33	902.13 ^3+^	902.03 ^3+^	H5N6F1S1-PROC	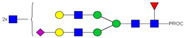
34	902.13 ^3+^	902.03 ^3+^	H5N6F1S1-PROC	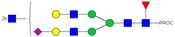
	863.77 ^3+^	863.68 ^3+^	H5N4F1S2-PROC	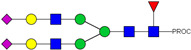
35	-	-	-	-
36	859.10 ^3+^	859.02 ^3+^	H6N6F1-PROC	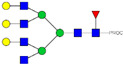
37	859.10 ^3+^	859.02 ^3+^	H6N6F1-PROC	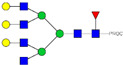
	931.43 ^3+^	931.37 ^3+^	H5N5F1S2-PROC	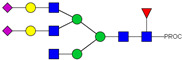
38	888.44 ^3+^	888.36 ^3+^	H6N5F1S1-PROC	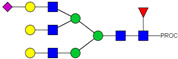
39	1051.97 ^2+^	1051.92 ^2+^	H9N2-PROC	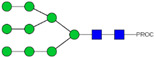
40	956.12 ^3+^	956.05 ^3+^	H6N6F1S1-PROC	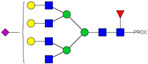
	999.15 ^3+^	999.06 ^3+^	H5N6F1S2-PROC	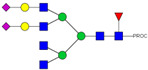
42	985.47 ^3+^	985.39 ^3+^	H6N5F1S2-PROC	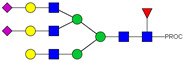
43	985.47 ^3+^	985.39 ^3+^	H6N5F1S2-PROC	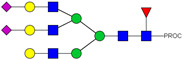
	913.12 ^3+^	913.04 ^3+^	H7N6F1-PROC	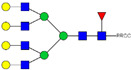
44, 45	790.16 ^4+^	790.06 ^4+^	H6N6F1S2-PROC	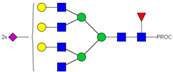
46	1010.14 ^3+^	1010.07 ^3+^	H7N6F1S1-PROC	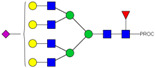
47	1082.48 ^3+^	1082.42 ^3+^	H6N5F1S3-PROC	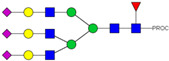
48	-	-	-	-
49	862.83 ^4+^	862.84 ^4+^	H6N6F1S3-PROC	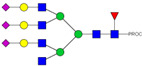
50, 51	830.66 ^4+^	830.58 ^4+^	H7N6F1S2-PROC	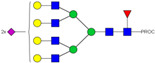
52, 53	903.44 ^4+^	903.35 ^4+^	H7N6F1S3-PROC	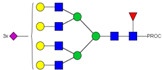
54, 55	976.19 ^4+^	976.12 ^4+^	H7N6F1S4-PROC	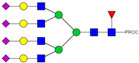

**Table 2 ijms-23-13455-t002:** Predicted Monosaccharide Compositions and Suggested Glycan Structures for Neu-IgE based on observed MS m/z masses. Numbered peaks are shown in Figure 3. H = Hexose; N = N-Acetylhexosamine; F = Fucose; S = Sialic Acid.

Peak Number	Observed MS	Calculated MS	Predicted Monosaccharide Composition	Suggested Structure
1	740.42 ^2+^	740.33 ^2+^	H3N3F1-PROC	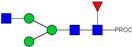
2, 3	841.95 ^2+^	841.87 ^2+^	H3N4F1-PROC	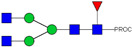
4, 5	727.90 ^2+^	727.81 ^2+^	H5N2-PROC	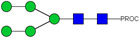
6	943.50 ^2+^	943.41 ^2+^	H3N5F1-PROC	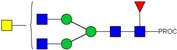
7	943.50 ^2+^	943.41 ^2+^	H3N5F1-PROC	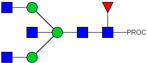
8	922.98 ^2+^	922.89 ^2+^	H4N4F1-PROC	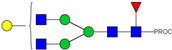
9	914.97 ^2+^	914.89 ^2+^	H3N4F2-PROC	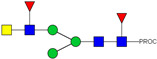
10	922.98 ^2+^	922.89 ^2+^	H4N4F1-PROC	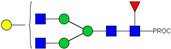
11	943.48 ^2+^	943.41 ^2+^	H3N5F1-PROC	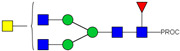
1045.05 ^2+^	1044.94 ^2+^	H3N6F1-PROC	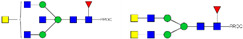
12	808.92 ^2+^	808.84 ^2+^	H6N2-PROC	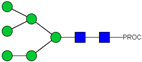
13	683.37 ^3+^	683.29 ^3+^	H4N5F1-PROC	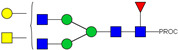
678.07 ^3+^	677.96 ^3+^	H3N5F2-PROC	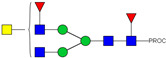
14	683.36 ^3+^	683.29 ^3+^	H4N5F1-PROC	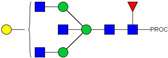
15	683.38 ^3+^	683.29 ^3+^	H4N5F1-PROC	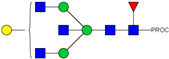
16	996.02 ^2+^	995.92 ^2+^	H4N4F2-PROC	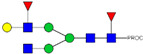
17	669.68 ^3+^	669.61 ^3+^	H5N4F1-PROC	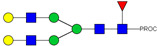
18	1024.48 ^2+^	1024.43 ^2+^	H4N5F1-PROC	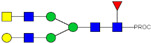
745.74 ^3+^	745.65 ^3+^	H3N6F2-PROC	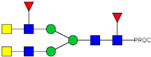
751.06 ^3+^	750.98 ^3+^	H4N6F1-PROC	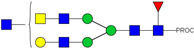
20, 21	889.94 ^2+^	889.86 ^2+^	H3N6F1S1-PROC	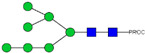
22, 23	737.40 ^3+^	737.31 ^3+^	H5N5F1-PROC	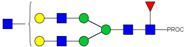
26, 27, 28	805.08 ^3+^	805.00 ^3+^	H5N6F1-PROC	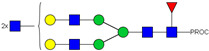
30	791.42 ^3+^	791.33 ^3+^	H5N5F1S1-PROC	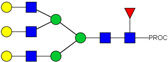
31	970.95 ^2+^	970.89 ^2+^	H8N2-PROC	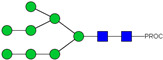
32	791.42 ^3+^	791.33 ^3+^	H5N5F1S1-PROC	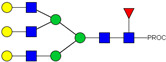
36, 37, 38	859.10 ^3+^	859.02 ^3+^	H6N6F1-PROC	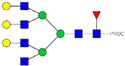
39	1051.97 ^2+^	1051.92 ^2+^	H9N2-PROC	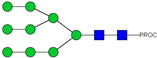
40, 41	926.78 ^3+^	926.71 ^3+^	H6N6F1S1-PROC	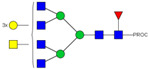
42, 43, 44	913.12 ^3+^	913.04 ^3+^	H7N6F1-PROC	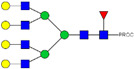

**Table 3 ijms-23-13455-t003:** Buffer recipes.

Buffer Name	Preparation
Buffer 1	50 mM Sodium Citrate + 50 mM Sodium Chloride, pH 3.5
Buffer 2	0.1 M Glycine, pH 2.3
Buffer 3	20 mM Citric Acid, pH 3.0
Neutralization Buffer	1 M Tris, pH 8.2
Calcium Buffer	0.1 mM CaCl_2_
FACS Buffer	1× HBSS, 2% FBS
Reducing Buffer	50 mM Dithiothreitol (DTT) in 4× Laemmli Protein Sample Buffer
T-PBS	0.1% Tween + PBS

## Data Availability

All data are contained within this manuscript.

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
