# Peer review of "Generation and Characterization of Native and Sialic Acid-Deficient IgE"

_ijms, 2022, doi:10.3390/ijms232113455_

Round 1

Reviewer 1 Report

In this manuscript, McCrew and colleagues provide a detailed methodological overview of IgE purification and follow-up quality assessment using a commercially available kit, CaptureSelectTM IgE Affinity Matrix (ThermoFisher). Their data provide sufficient evidence and justifications that the CaptureSelectTM I IgE-specific resin columns provide an efficient pipeline for purification of ‘highly pure’ recombinant IgEs for use in AllergoOncology R&D, compared to conventional methods. They further provide a comprehensive analysis of desialylated IgE, which may have applications in basic and translational research and additionally guide the development of future glycol-engineered IgE-based therapeutics.

Overall, this is a well-written manuscript and I have no major concerns. Although, I would have liked to see some functional assessment of the native vs. desialylated IgEs too, eg, IgE-dependent cytotoxic assays.

Please find my recommendations below:

1-      Fig 3: Treatment of IgE samples with neuoaminidase-A is expected to produce desialylated IgEs with slightly lower molecular weights than the native IgEs. This does not seem to be reflected in Figure 3C-D?

2-      Line 232-233: the sentence requires grammatical revision.

3-      Fig 4: Text size should be increased for the figure labels. Furthermore, the current graph title (A2058) in Fig4C should be amended to CSPG4 to reflect the antigen.

4-      Figure 6: As above, some of the labels are currently too small and should be scaled up accordingly.

Reviewer 2 Report

The manuscript by McCraw et al describes the use of Capture Select IgE matrix for the purification of chimeric IgE, before or after removing its sialic acid. Overall, the manuscript is well-written, the flow and the results are presented adequately.

Yet, I have several major concerns about the scientific novelty of this manuscript.

(1)    The authors claim the there is “a growing demand for the generation, purification and study of recombinant IgE…” (line 54). However, they do not provide ant references to back this statement. Moreover, the use of commercial matrices, such as the products that were applied in the manuscript, are well-established and have been used for many years for the purification of IgE from different sources (recombinant or native).

(2)    Basically, the authors have compared a well-established matrices for the purification of IgE antibodies using the manufacturer’s recommended setting. While the authors declare that they have “optimized” the pipeline (lines 304 and 375) and present “improved purification” (line 305), it is not what is the optimized protocol. For example, the elution buffer which provided the best yield, is actually the one recommended by the manufacturer.

(3)    The use of Neu-A to demonstrate the purification of desialyated IgE, although performed well, cannot represent the general purification of “altered glycan profile” (lines 190-191).

Therefore, while manuscript do provides important technical data regarding IgE purification, especially in small-scale volumes and will be of interest to the antibody research community, I believe it will be more suitable for publication in a methods/protocols oriented journal.

Reviewer 3 Report

This manuscript by McCraw et al studied the optimization of human IgE purification methods based on 2 of the IgEs (CSPG-4 and HER2) published in 2018 and 2019. The paper is well written, supported by some of the relevant experiment and detailed discussion, as a straightforward technical report.

General comments:

1.     My main concern is with the SDS-PAGE, HPLC and FACS analysis, there is nothing showing whether these IgEs are bioactive? Therefore representing only a minor increment in knowledge, this is reflected in the conclusion (Pg 20, line 397-404).

2.     Fig 5B, CD23, while there is a clear shift in terms of CD23 FITC expression with Neu-IgE, the MFI difference is at best 30%, ie, not even one log when compared to the control IgE, and the control CSPG-4 IgE is there as an isotype control? More detail explanation would be informative.

3.     Aside, form FACS, can these purified IgE antibodies be employed in other in vitro techniques, or at least in immunohistochemistry or ELISA?

Minor comment:

Please consider Fig 6 as supplementary figure.

Reviewer 4 Report

This is an excellent straightforward study of artificial IgE production. The authors established a production and purification strategy of native human IgE and a glyco-engineered human IgE variant with low sialic acid.  The results could provide an efficient pipeline for the purification of recombinant IgEs for clinical use. 

Round 2

Reviewer 2 Report

I believe that this manuscript will be more suitable for publication in a methods/protocols oriented journal.

Reviewer 3 Report

The authors have addressed all the comments, with new Fig5D (degranulation assay) and revision on Fig5A-C.